# Development of ARCADIA: a tool for assessing the quality of peer-review reports in biomedical research

Cecilia Superchi ![ORCID],[1,2] Darko Hren ![ORCID],[3] David Blanco ![ORCID],[1,2] Roser Rius,[2] Alessandro Recchioni,[4] Isabelle Boutron,[1,5] José Antonio González[2]

IB and JAG contributed equally.

[1]Université de Paris, CRESS, INSERM, INRA, Paris, France
[2]Statistics and Operations Research Department, Barcelona-Tech, UPC, Barcelona, Spain
[3]Department of Psychology, School of Humanities and Social Sciences, University of Split, Split, Croatia
[4]Springer Nature, Berlin, Germany
[5]Centre d'épidémiologie Clinique, Hôpital Hôtel-Dieu, Assistance Publique Hôpitaux de Paris (APHP), Paris, France

**Correspondence to**
Cecilia Superchi;
cecilia.superchi@upc.edu

## ABSTRACT

**Objective** To develop a tool to assess the quality of peer-review reports in biomedical research.

**Methods** We conducted an online survey intended for biomedical editors and authors. The survey aimed to (1) determine if participants endorse the proposed definition of peer-review report quality; (2) identify the most important items to include in the final version of the tool and (3) identify any missing items. Participants rated on a 5-point scale whether an item should be included in the tool and they were also invited to comment on the importance and wording of each item. Principal component analysis was performed to examine items redundancy and a general inductive approach was used for qualitative data analysis.

**Results** A total of 446 biomedical editors and authors participated in the survey. Participants were mainly male (65.9%), middle-aged (mean=50.3, SD=13) and with PhD degrees (56.4%). The majority of participants (84%) agreed on the definition of peer-review report quality we proposed. The 20 initial items included in the survey questionnaire were generally highly rated with a mean score ranging from 3.38 (SD=1.13) to 4.60 (SD=0.69) (scale 1–5). Participants suggested 13 items that were not included in the initial list of items. A steering committee composed of five members with different expertise discussed the selection of items to include in the final version of the tool. The final checklist includes 14 items encompassed in five domains (Importance of the study, Robustness of the study methods, Interpretation and discussion of the study results, Reporting and transparency of the manuscript, Characteristics of peer reviewer's comments).

**Conclusion** Assessment of Review reports with a Checklist Available to eDItors and Authors tool could be used regularly by editors to evaluate the reviewers' work, and also as an outcome when evaluating interventions to improve the peer-review process.

## BACKGROUND

Editorial peer review stands as the gateway to scientific publication. The process was established to ensure that research papers are vetted by independent experts before they are published, although it is recently being increasingly questioned due to beliefs that it is flawed.[1 2] Despite efforts over the last

### Strengths and limitations of this study

► Assessment of Review reports with a Checklist Available to eDItors and Authors (ARCADIA) constitutes the first tool that has been systematically developed to assess the quality of peer-review reports.
► Its development is based on an exhaustive review of the literature and on empirical data from a large and heterogeneous sample of both biomedical editors and authors.
► The majority of editors and authors were from Europe and North America, which may limit the generalisability of the results.
► ARCADIA has not yet been validated.

30 years to 'make peer-review scientific', its impact is still considered suboptimal.[3]

Peer reviewers, who are the pivotal actors in this process, are requested to write a review report evaluating the submitted manuscript. A peer-review report helps authors to improve the quality of their manuscripts, and it also helps editors make an informed decision about the outcome of the manuscript. However, evidence shows that these peer-review reports are often of poor quality.[4 5]

Tools for assessing the quality of peer-review reports have been proposed, of which we have conducted a systematic review and identified 24 tools: 23 scales and 1 checklist.[6] However, none reported any definition of peer-review report quality, only one described the scale development, and 10 provided measures of reliability and validity. Further, the development and validation process resulted from a small consensus of people, and the concepts evaluated by these tools were quite heterogeneous.

In 2016, Bruce *et al* published a review evaluating the impact of interventions to improve the quality of the peer-review process.[5] The authors showed that it is essential to clarify the outcomes (such as, the quality of peer-review

reports), which should be used in randomised controlled trials to evaluate these interventions.

A validated tool is direly needed to clearly define the quality of a peer-review report in biomedical research. This tool could be used regularly by editors to evaluate the reviewers' work, and also as an outcome when evaluating interventions to improve the peer-review process. In the present study, we report on the development of a new tool to assess peer-review reports in biomedical research.

## METHODS

### Steering Committee

We formed a steering committee of five members (CS, DH, AR, IB and JAG), whose expertise include clinical epidemiology, biostatistics, social science and editorial peer review. The steering committee agreed on how to define peer-review report quality; they agreed on the survey questionnaire based on the results of a previous systematic review[6]; they interpreted the results of the survey and they agreed on the final version of the tool.

### Defining the tool's objective

The tool aims to assess the quality of peer-review reports in biomedical research. We defined the quality of a peer-review report as 'the extent to which a peer-review report helps editors make a fair decision and authors improve the quality of the submitted manuscript'.

### Generating the items

A systematic review allowed the identification of 24 tools, aimed at assessing the quality of peer-review reports.[6] We extracted 132 items from such tools. After removing the redundant items, we obtained 17 items. We then eliminated two items and incorporated five new ones that met our definition of peer-review report quality, after piloting the survey questionnaire and discussing with the steering committee. Overall, 20 items were identified to assess peer-review report quality (table 1).

### Survey

We conducted an online survey of editors and authors in order to: (1) determine if they endorse the proposed definition of peer-review report quality; (2) identify the most important items to include in the final tool and (3) identify any new items that should be included.

### Survey questionnaire

The questionnaire was constructed using the online survey software SurveyMonkey.[7] It was structured into four main parts and included both open and multiple-choice questions. First, the participants were asked to agree ('yes/no/partially') on the definition we provided for peer-review report quality. They were also invited to add any comments or ideas on how to improve the definition. Second, they were asked to rate the importance of the 20 items for assessing the quality of peer-review reports we identified. Their responses were based on a 1–5 Likert

**Table 1** The 20 items to assess peer-review (PR) report quality included in the survey

| Labels | Items to assess PR report quality |
|---|---|
| Relevance | The reviewer comments on the relevance of the study |
| Originality | The reviewer comments on the originality of the study |
| Interpretation results | The reviewer comments on the interpretation of study results |
| Strengths and weaknesses (general) | The reviewer comments on the general strengths and weaknesses of the study |
| Strengths and weaknesses (methods) | The reviewer comments on the strengths and weaknesses of the study methods |
| Statistical methods | The reviewer comments on the appropriateness of the statistical methods |
| Methodological quality | The reviewer comments on the methodological quality (internal validity) of the study |
| Applicability and external validity | The reviewer comments on the applicability and external validity of the study results |
| Presentation and organisation | The reviewer comments on the presentation and organisation of the manuscript |
| Adherence to reporting guideline (RG) | The reviewer comments on the adherence of the manuscript to the reporting guideline |
| Structure of reviewer's comms. | The reviewer's comments are structured and organised |
| Clarity | The reviewer's comments are clear and easy to read |
| Constructiveness | The reviewer's comments are constructive |
| Detail/Thoroughness | The reviewer's comments are detailed and thorough |
| Objectivity | The reviewer's comments are objective |
| Fairness | The reviewer's comments are fair |
| Support by evidence | The reviewer's comments are evidence based |
| Knowledgeability | The reviewer knows and understands correctly the content of the manuscript |
| Tone | The reviewer uses a courteous tone |
| Timeliness | The reviewer completes the PR report on time |

scale (1 being not important and 5 very important). In particular, we asked the participants if the item should be included in a tool for assessing the quality of peer-review reports. Moreover, they were invited to comment on the importance and wording of each item. In order to eliminate the question order effect, the items appeared in random order for each respondent. Third, the participants were invited to suggest any additional items missing that they considered important for assessing the quality of peer-review reports. Finally, the questionnaire included nine demographic questions related to sex, age, education level, job title, referring institution and job experience as biomedical editor and/or author. We developed two versions of the questionnaire because biomedical editors and authors were recruited differently, despite the fact that some of them could play both roles (see online supplementary file 1). The two versions were structured in the same way; they only differed in some questions related to the demographic characteristics. The questionnaire was piloted among six experienced scientific editors and authors, followed by a subsequent revision based on their feedback.

### Participants and recruitment strategy
We targeted biomedical editors and authors using a purposive sampling approach to recruit a heterogeneous sample of information-rich cases.[8]

#### Biomedical editors
By means of standardised email, we invited two groups of editors to participate in the survey: 586 biomedical editors from 43 journals in the BMJ Publishing group; and 478 editors from 235 journals identified in a previous cross-sectional bibliometric study[9] (see online supplementary file 2). The survey was also distributed to 27 editors from 48 journals in BMC (part of Springer Nature), using internal email and to members of the European Association of Science Editors (EASE) through their newsletter. In the invitation email and newsletter, the editors were encouraged to forward the survey to colleagues who might be interested in issues related to peer-review. This recruitment strategy, known as snowballing, allowed us to identify 'information-rich key informants' among biomedical editors.[8] On the first page of the survey, participants were informed that the collected data would be anonymous, and they were further asked if they would agree to share their deidentified data in an open access repository. Two reminder emails were sent to non-respondents. Finally, the survey was promoted on Twitter and on the EASE blog[10] and Methods in Research on Research[11] websites.

#### Authors
Searching the top 30-biomedical journals with the highest impact factors, we identified 4396 corresponding authors of articles that reported original research and which were published in Medline between 1 February and 31 October 2018 (see online supplementary file 3). We used the R package easy PubMed to extract the email contacts.[12] The

corresponding authors received a standardised email that explained the purpose of the study and included a link to the survey (see online supplementary file 2). The first page of the survey informed participants that the data were collected anonymously and also asked if they would agree to share their deidentified data in an open-access repository. Two reminder emails were sent to non-respondents.

We did not use a snowballing strategy to recruit authors. However, since the survey directed to biomedical editors was promoted on Twitter by different users who sometimes did not provide thorough instructions, we included in the first page of the survey, also the link to the questionnaire addressed to authors. This was done so that a researcher, who was not an editor and mistakenly opened the link to the survey questionnaire, was still able to participate to the study as biomedical author.

### Data analysis
We described the demographic data in terms of frequencies and percentages. The importance of the 20 items to assess peer-review report quality is described in means and proportions of editors or authors who rated the importance of the items from 1 to 5. The items were also sorted according to the mean raking of all participants and either editors or authors. We also calculated Pearson correlations among items. The calculations and graphical representations were all obtained using the statistical software R 3.5.0.[13]

#### Principal component analysis of quantitative data
We conducted a principal component analysis (PCA) to examine item redundancy among the 20 items to assess peer-review report included in the survey. PCA is a multivariate statistical technique used to reduce the number of variables in a dataset to a smaller number of dimensions.[14] The new dimensions (or principal components (PC)) are mutually independent and are determined by choosing the directions that explain the most variation in the data. The first PC (PC1) accounts for the largest possible variance in the data, and each succeeding PC accounts for decreasing amounts of the remaining. This exploratory analysis helps reveal simple underlying structures in complex datasets. We performed PCA using the R package FactoMineR.[15]

#### Inductive content analysis of qualitative data
We used a general inductive approach for qualitative data analysis. In particular, we followed the five steps of inductive analysis proposed by David R. Thomas: (1) Preparation of raw data files; (2) Close reading of text; (3) Creation of codes; (4) Overlapping coding and uncoded text and (5) Continuing revision and refinement of themes system.[16] In the third phase, two investigators (CS and DB) created independently the initial codes from the responses of the first 100 participants for each open-ended question. In order to ensure consistency and credibility, the initial codes were discussed with a third

investigator (DH) and a codebook was developed and was used for analysing the remaining responses. In case new codes were successively created from the remaining responses, the emerging codes were added to the codebook and applied to entire dataset. Two investigators (CS and DH) reviewed and refined the codebook and further clustered the codes into major themes. We used the software NVivo V.12 for data management and analysis.[17]

### Selecting items
The steering committee reviewed all items and, ultimately, drafted and refined the final version of the tool. Based on the participants' qualitative and quantitative answers, redundant items were combined, existing items were modified and/or expanded on, and new items proposed by survey participants were added.

### Patient or public involvement
Patients and members of the public were not involved in the study.

## RESULTS
### Participants
Between 7 November 2018 and 4 February 2019, 198 biomedical editors and 248 authors completed the survey. Of the 1134-biomedical editors and 3633 corresponding authors invited via email, 89 (7.8%) and 238 (6.5%) completed the survey, respectively. In addition, 109 editors and 10 authors completed the survey using the web link.

Participants were mainly male (263/399, 65.9%) with a PhD degree (225/399, 56.4%), and their ages were equally distributed across ranges (mean=50.3, SD=13). They were mainly located in Europe (219/389, 56.3%) and North America (118/389, 30.3%). More than half of the editors had work experience of more than 5 years (91/165, 55.2%), while over one-third of the authors had work experience of more than 20 years (84/224, 37.5%) (see table 2). Editors were mainly associate editors (63/165, 38.2%) and editors in chief (50/165, 30.3%), primarily involved in making decisions on the submitted manuscripts (144/165, 87.3%). Most of them worked in specialty journals (126/165, 76.4%) and they were used to contribute as authors in scientific papers (141/165, 85.5%). The corresponding authors were mainly professors (63/224, 28.1%), but also PhD students, postdocs or lecturers (49/224, 21.9%) or researchers (47/224, 21%). The majority of them worked in public universities (134/224, 59.8%) and they were not employed as editor (161/224, 71.9%) in biomedical journals. Among those who also work as biomedical editors (63/224, 28.1%), 88.9% of them are involved in making decision on the manuscript (online supplementary file 4).

### Definition of peer-review report quality
Overall 84% (362/431) participants, precisely 85% (160/188) editors and 83% (202/243) authors, agreed on the definition of peer-review report quality that we provided in the survey. The definition was slightly modified to take into account participants comments (online supplementary file 5). The quality of a peer-review report is now defined as 'the extent to which a peer-review report helps, first, editors make an informed and unbiased decision about the manuscripts' outcome and, second, authors improve the quality of the submitted manuscript'.

### Quantitative results
We created a web application that is publicly available at https://www-eio.upc.edu/redir/ReportQuality. Through the application, the readers can easily access and explore the quantitative results of the survey.

### Rating the importance of items
The items were generally highly rated, with a mean score ranging from 3.38 (SD=1.13) to 4.60 (SD=0.69). All the items were scored 4 or 5 by >50% of the participants (see web application). The three items rated as the most important were: (1) *Knowledgeability*; (2) *Methodological quality* and (3) *Fairness*. The three least important items were: (1) *Originality*, (2) *Presentation and organisation* and (3) *Adherence to RG*.

A peer-review report aims to help authors improve their submitted manuscripts and assist editors in taking editorial decisions. Due to this dual objective, we compared editors' and authors' mean scores in order to investigate whether any difference is found in their perceptions regarding the importance of the 20 items that assess peer-review report quality. We found little discrepancy in the mean scores between biomedical editors and authors, with only two items indicating any difference: (1) *Timeliness* and 2) *Detail/Thoroughness*. The *Timeliness* of the peer-review report was considered more important to authors than to editors (respectively, in the 12th and 16th rank positions). Meanwhile, editors rated the *Detail/Thoroughness* of the reviewer's comments higher than did authors (respectively, in the 11th and 16th rank positions).

### Correlations among items
Overall, we found relatively weak positive correlations among items. The largest positive correlations were found between *Relevance* and *Originality*, and between *Fairness* and *Objectivity* (r=0.55 and 0.43, respectively).

### Principal component analysis
The first PC1 accounted for 22.1% of data variability. The next two dimensions (PC2 and PC3) accounted for 38.5% of the cumulative variability and contributed gradually, that is, they increased at only small increments. PC1 was positively correlated to all items (or variables), and it showed correlations higher than 0.4—which is the figure commonly used as a threshold reference for factor loadings—for 16 out of 20 items (see web application). These results illustrate that the data variance was not concentrated in a few components but distributed across all of them; hence, reducing the number of items is not

**Table 2** Survey participants' characteristics

| Characteristics | Editors n=198 | Authors n=248 | Total n=446 |
|---|---|---|---|
| **Gender** | **n=169 (%)** | **n=230 (%)** | **n=399 (%)** |
| Woman | 46 (27.2) | 83 (36.1) | 129 (32.3) |
| Man | 121 (71.6) | 142 (61.7) | 263 (65.9) |
| Other | 2 (1.2) | 5 (2.2) | 7 (1.8) |
| **Age** | **n=156 (%)** | **n=220 (%)** | **n=376 (%)** |
| <40 | 32 (20.5) | 71 (32.3) | 103 (27.4) |
| 41–50 | 29 (18.6) | 59 (26.8) | 88 (23.4) |
| 51–60 | 52 (33.3) | 37 (16.8) | 89 (23.7) |
| >60 | 43 (27.6) | 53 (24.1) | 96 (25.5) |
| **Education** | **n=169 (%)** | **n=230 (%)** | **n=399 (%)** |
| Bachelor degree | 4 (2.4) | 3 (1.3) | 7 (1.7) |
| Master degree | 11 (6.5) | 20 (8.7) | 31 (7.8) |
| PhD | 107 (63.3) | 118 (51.3) | 225 (56.4) |
| MD or equivalent | 34 (20.1) | 76 (33.0) | 110 (27.6) |
| Prefer not to answer | 2 (1.2) | 1 (0.4) | 3 (0.7) |
| Other | 11 (6.5) | 12 (5.2) | 23 (5.8) |
| **Location journal/institution** | **n=165 (%)** | **n=224 (%)** | **n=389 (%)** |
| Europe | 132 (80.0) | 87 (38.8) | 219 (56.3) |
| North America | 23 (14.0) | 95 (42.4) | 118 (30.3) |
| South America | 2 (1.2) | 5 (2.2) | 7 (1.8) |
| Africa | 1 (0.6) | 1 (0.4) | 2 (0.5) |
| Asia | 3 (1.8) | 11 (5.0) | 14 (3.6) |
| Australia | 4 (2.4) | 25 (11.2) | 29 (7.5) |
| **No of years of experience** | **n=165 (%)** | **n=224 (%)** | **n=389 (%)** |
| <5 years | 74 (44.8) | 36 (16.1) | 110 (28.3) |
| 6–10 years | 46 (27.9) | 51 (22.7) | 97 (24.9) |
| 11–15 years | 27 (16.4) | 34 (15.2) | 61 (15.7) |
| 16–20 years | 7 (4.2) | 19 (8.5) | 26 (6.7) |
| >20 years | 11 (6.7) | 84 (37.5) | 95 (24.4) |

recommended, since this would imply an important loss of data information.

The study of the supplementary variables did not reveal any differences between authors and editors in terms of items rating. However, we found that female participants above the age of 55 years old generally provided higher rating for the items, compared with younger male participants.

### Qualitative results
#### Comments on importance and/or wording of items
Out of 446 survey participants, 267 (59.9 %) made at least one comment on the importance and/or wording of the items. Based on the initial coding of the comments, we were able to identify eight general themes that they addressed: Peer reviewer; Wording; Importance; Dependency; Responsibility; Item; Structure and content; and Improvement. Table 3 reports the eight themes together with their definition and the most frequent codes (n>5), with example quotes. The entire codebook is found in online supplementary file 5.

#### New items
Participants suggested 13 items that were not included in the initial list of items. These items are listed in online supplementary file 6. The entire codebook is found in online supplementary file 5.

### Steering committee meeting
The steering committee met on the 19 July 2019 to discuss the selection of items to include in the final version of the tool. Their decisions were based on the participants' quantitative and qualitative answers. The flow of the items is summarised in figure 1.

The items *Relevance* and *Originality* were merged into a new item named *Contribution* (of the study). This decision

**Table 3** Survey participants' comments on the importance and/or wording of the 20 items to assess peer-review report quality

| Themes | Definition | Codes | Examples |
|---|---|---|---|
| Dependencies | Theme including codes on how the importance of an item depends on different factors (e.g., type of study, paper quality, type of journal, etc.) | Dependency on the type of study (n=34) | Depends on type of study. For systematic reviews of course fundamental. For other studies this will be more and more important for easier comparisons between studies and for quality improvement. It makes our work easier if the authors'compliance also improve. |
| | | Dependency on the paper quality (n=20) | This depends on the quality of the manuscript. Sometimes the quality is so low that a reviewer can highlight one or two major methodological flaws, which are sufficient to reject. |
| | | Dependency on the type of journal (n=19) | This depends on the journal's criteria. |
| | | Dependency on the author's claim and impact of the study (n=7) | This depends on the claims made. |
| Importance | Theme including codes on the importance (or not) of an item. | Importance of the item (n=43) | This is absolutely key to the interpretation of the study. Unfortunately most reviewers, in my field, do not fully understand current (and correct) methods. |
| | | Importance of replication and conformation study (n=18) | Not always important to be original study as some are trying to duplicate findings from previous studies. |
| | | Importance of perceptions, opinions and experience (n=14) | Some comments will inevitably be opinion, regarding emphasis, values, writing style. |
| | | Importance of a high-quality review rather than on time review (n=13) | Better to have a late high quality report than a moderate quality report on time. |
| Improvements | Theme including codes on how an item is useful for both authors and editors in the peer-review process. | Useful for authors and editors (n=21) | It's important to make it easy for the editor and authors to understand the review, and for authors to respond. |
| | | Improving the manuscript (n=9) | Important when it will help improve the quality of the communication. Not necessary when it flows well. |
| | | Avoiding exaggeration and misinterpretation (n=8) | This is an area where the reviewer may have a valuable role in tempering an author's enthusiasm, hubris or bias. |
| Item | Theme including codes on the characteristics of an item. | Related to other item (n=43) | Yes, but it is confusing to separate this from the general strength and weaknesses. The question should be if the reviewer thinks that the message can (potentially) answer the research question. |
| | | Subjective item (n=22) | Too subjective! What is relevant to one person of field could be totally not-relevant to another. |
| | | Requirement (n=9) | It's an ethical requirement, and helps improve everyone's experience. |
| Reviewer | Theme including codes on the expertise and characteristics of a peer reviewer. | Reviewer's expertise (n=148) | Some reviewers know about methods and some about content. It would be ideal to always have both, but that is often not the case. |
| | | Impossibility to be totally objective (n=35) | 100% objectivity doesn't exist. |
| | | Reviewer as an extra unpaid job (n=10) | For the most part, reviews are done on a voluntary basis. |
| Responsibility | Theme including codes on the editor and/or author's responsibility to assess an item. | Editor's responsibility (n=48) | In my experience this is usually picked up by the Editors and Associate Editors rather than the reviewers. |
| | | Joint responsibility (n=24) | I think this is the role of the editors as well as the reviewers. |
| | | Author's responsibility (n=6) | Authors should already be doing this. |

**Table 3** Continued

| Themes | Definition | Codes | Examples |
|---|---|---|---|
| Structure and content | Theme including codes on the structure and content of a peer-review report. | Straight to the critical points (n=14) | Sometimes a succinct review is still helpful, if it cuts straight to the critical points. For example, if it is clear that a manuscript has major flaws, then a review that points out those flaws clearly and dispassionately would be very helpful. It would not necessarily need to delve into the finer details. |
| | | Unnecessary to provide evidence to each comment (n=10) | I don't think reviewers need to cite something for every point that they make. |
| | | Declaration of COI (n=8) | Peer reviewers should disclose COI. |
| | | Standard structure of a review (n=7) | I would suggest providing a template to reviewers. |
| | | Not necessary for all reviews (n=6) | Reviews come in all lengths and vary in detail. It is helpful to have some reviewers provide detailed information but not necessary that all do so. |
| Wording | Theme including codes on how to improve the wording of an item. | Wording of the item (n=110) | Rather than 'The reviewer's comments are evidence-based' I would suggest that the category should be: 'The reviewer distinguishes between comments that are supported by evidence (and provides suitable citations) and those based on opinion or experience'. |

was based on the high positive correlation found between the two items (0.55) and on the participants' opinions. Furthermore, participants suggested in their comments that the item *Relevance* was '*highly subjective*', because '*each reviewer's decision on relevance reflects what is relevant to them, which may not reflect relevance to the journal*'. They also believed that the *Originality* of a study is not always an important aspect for comments in a peer-review report, because some manuscripts '*are trying to duplicate findings from previous studies*'. They, therefore, suggested reformulating the two items by asking the reviewer what the study '*adds to our knowledge*'.

The steering committee decided to include the item *Interpretation of results* as a domain of the tool instead of a single item, changing the name into *Interpretation and discussion of the study results*. This decision resulted from the addition of two new items (*Study conclusions* and *Study limitations*), based on the suggestions of survey participants. The domain *Interpretation and discussion of the study results* now encompasses three items: (1) *Study conclusions*; (2) *Study limitations* and (3) *Applicability and generalisability*.

Overall, survey participants believed that the items *Strengths and weaknesses (general)* and *Strengths and weaknesses (methods)* were '*confusing to separate*'. Additionally, the steering committee agreed that *Strengths and weaknesses (methods)* and *Methodological quality* were also redundant; thus, it was ultimately decided to merge the three items into a new item named *Study methods*.

The items *Objectivity* and *Fairness* were merged because of both the moderate correlation between them (0.43) and the participants' opinions. Participants suggested that the total objectivity of the reviewer's comments is not possible because '*all decisions contain some personal biases and subjectivity*' and they also believed that the term

fairness was '*very subjective*' and difficult to define. Additionally, the steering committee agreed to also combine these two items into *Supported by evidence*. The committee finally decided to merge all three items into *Objectivity*, and this was defined as 'comments provided in a peer-review report should be as objective as possible and, if considered appropriate, include references to support the reviewer's statements'.

The steering committee agreed to merge *Structure of reviewer's comments* and *Clarity*, because participants considered both important for making the peer-review report easy '*to read for both editors and authors*'. Moreover, participants suggested that the *Detail/Thoroughness* of a peer-review report was mostly associated with the quality of a manuscript, because in certain occasions a study can be so poorly conducted that '*a reviewer can highlight one or two major methodological flaws*' without conducting a detailed review. They, therefore, believed that a detailed report is not '*always necessary*' and instead preferred a succinct report that '*cuts straight to the critical points*'. Taking into account the participants' opinions, the steering committee finally decided to include a single item named *Clarity*, which is defined as 'a peer-review report should be clear, succinct and well organised in order to be understood correctly by editors and authors'.

The items *Tone* and *Constructiveness* were merged into *Constructiveness*, which is defined as 'a peer-review report should contain constructive and polite comments that allow the authors to improve the quality of their work'. This decision was based on the participants' opinions that '*the comments should be polite and constructive*'.

The item *Adherence to RG* and the new item *Reproducibility* suggested by survey participants were merged into *Reporting* based on the steering committee decision.

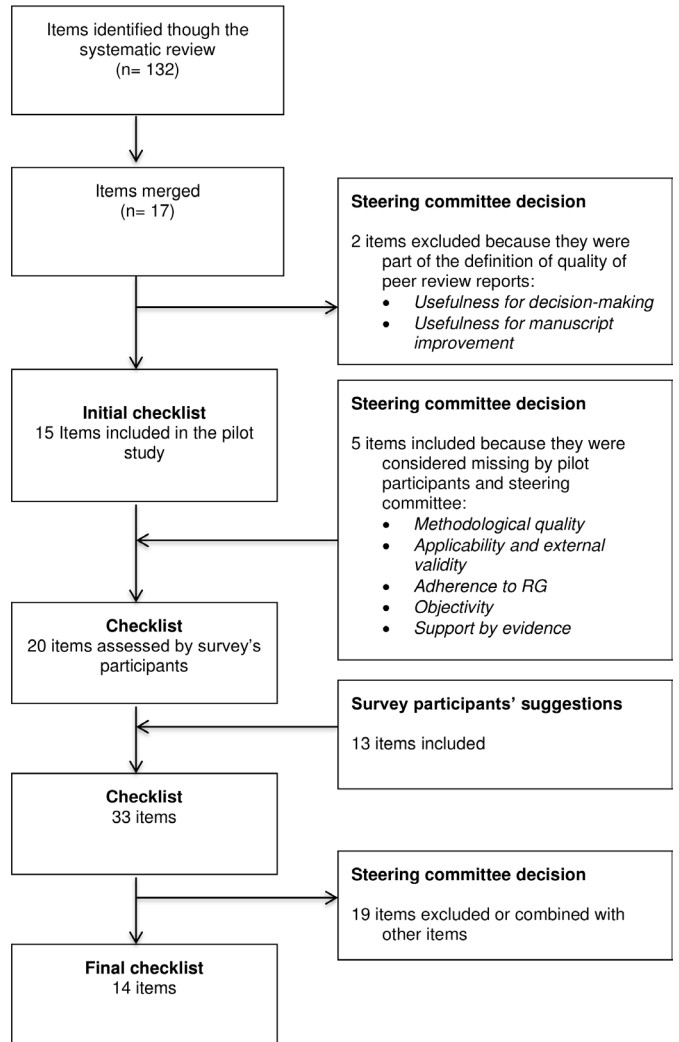

**Figure 1** Flow chart of items to include in a checklist to assess the quality of peer-review reports.

The item *Reporting* was defined as 'the reviewer should comment if the reporting of the study is clear, complete and transparent enough for facilitating its reproducibility by verifying the adherence of the manuscript to the corresponding reporting guideline'.

The items *Timeliness* and *Knowledgeability* were not included in the final version of the tool. Survey participants suggested that *Timeliness* was not '*directly tied to review quality*' because '*some of the best reviews come in past the deadline*'. Furthermore, the steering committee agreed that the item *Knowledgeability* was generally difficult to assess, because it implied that anyone using the tool would have enough competence to evaluate the reviewer's knowledge and expertise. Five new items suggested by survey participants (*Data availability, Study protocol, Study conclusions, Study limitations* and *Relevant literature*) were finally included in the tool.

## The Assessment of Review reports with a Checklist Available to eDItors and Authors tool

The Assessment of Review reports with a Checklist Available to eDItors and Authors (ARCADIA) tool was finally

developed. The tool is a checklist that includes five domains and 14 items (table 4). Brief explanations of the items included in the five domains are provided in online supplementary file 7.

## DISCUSSION

This study resulted in a checklist of items to assess the quality of peer-review reports in biomedical research. The checklist constitutes the first tool that has been systematically developed to assess the quality of peer-review reports.

The checklist is simple, applicable to any biomedical field, and consists of five domains covering 14 items, each of which is phrased as a question. Each item should be ticked as 'yes' or 'no'. An item could be also checked 'NA' if it is not covered in the study (e.g., there are no data or other materials attached to the manuscript) and/or the peer reviewer is not qualified to comment on that specific aspect (e.g., statistical methods). The ARCADIA tool has several strengths. It is the first tool ever developed based on an exhaustive review of the literature[6] and on empirical data from a large sample of both biomedical editors and authors. Further, it is the only tool that clearly defines the quality of peer-review reports, as its definition was based on the perspectives of 446 authors and editors.

To develop the tool, we recruited a large sample of biomedical editors and authors with varying experience and backgrounds. We found the percentage of female participants who took part in the survey to be quite low (129/399, 32.3%). This is in line with evidence showing that gender equity in academic medicine careers remains far behind.[18] Moreover, we recruited corresponding authors (who are usually first authors) from the top 30 biomedical journals. Evidence also shows that women are under-represented as first authors among biomedical journals with high impact factors.[19]

Overall, we did not find any differences between authors and editors in terms of item rating by conducting PCA. Only two items, *Timeliness* and *Detail/thoroughness,* presented a difference according to the separate mean score rankings of authors and editors. Timeliness was considered more important for authors and this could be justified by the fact that authors are usually more interested in receiving decisions about their manuscript as soon as possible. Whereas, editors rated detail/thoroughness as more important to them, given thorough and detailed peer-review reports help them make a better editorial decision on any given manuscript.

The present study also has some limitations. The survey questionnaire included some open-ended questions, which allowed participants to voluntarily express their opinions. However, we were not able to inquire further to clarify and verify some information provided by the study's participants. Therefore, the interpretation of some information could be affected by the perception of the three investigators who conducted the qualitative analysis. Additionally, since participants could comment voluntarily on the importance and wording of each item,

**Table 4**  The ARCADIA tool

**In the peer review report, did the reviewer comment on…**

| | | |
|---|---|---|
| **Importance of the study** | the contribution of the study to scientific knowledge? | ☐ YES ☐ NO ☐ NA |
| | whether the relevant literature was accurately reviewed? | ☐ YES ☐ NO ☐ NA |
| **Robustness of the study methods** | the soundness of the study methods (e.g., study design, outcomes, risk of bias)? | ☐ YES ☐ NO ☐ NA |
| | the suitability of the statistical methods? | ☐ YES ☐ NO ☐ NA |
| **Interpretation and discussion of the study results** | whether the study conclusions answer the research question(s) and correctly summarise the study results? | ☐ YES ☐ NO ☐ NA |
| | whether the study limitations are acknowledged? | ☐ YES ☐ NO ☐ NA |
| | the applicability and generalisability (external validity) of the study results? | ☐ YES ☐ NO ☐ NA |
| **Reporting and transparency of the manuscript** | whether any major deviations from the study protocol are reported? | ☐ YES ☐ NO ☐ NA |
| | whether the completeness of the reporting allows study reproducibility, by verifying the adherence of the manuscript to the corresponding reporting guideline (RG)? | ☐ YES ☐ NO ☐ NA |
| | the presentation (e.g., quality of the written language, tables, figures, etc.) and organisation of the manuscript? | ☐ YES ☐ NO ☐ NA |
| | the availability of study data and material? | ☐ YES ☐ NO ☐ NA |
| **Were the peer reviewer's comments…** | | |
| **Characteristics of peer reviewer's comments** | clear? | ☐ YES ☐ NO |
| | constructive? | ☐ YES ☐ NO |
| | objective and, if opportune, supported by evidence? | ☐ YES ☐ NO |

ARCADIA, Assessment of Review reports with a Checklist Available to eDItors and Authors; NA, Not applicable.

the number of comments among items differed greatly. Furthermore, the majority of editors and authors were from Europe and North America, which may limit the generalisability of the results. This result may be due to the recruitment strategy we used, especially to identify biomedical editors. Although we also used a snowballing strategy, we mainly contacted editors through European biomedical journals. Finally, the present study reports on the first version of the ARCADIA tool, which has not yet been validated.

### Implications
The tool is a general checklist available to all biomedical editors and authors. It could be regularly used by editors to evaluate the reviewers' work, and it can also be used as an outcome when evaluating interventions in order to improve the peer-review process.

### CONCLUSIONS
ARCADIA is the first checklist that has been systematically developed to assess the quality of peer-review reports. It is based on the perspectives of a large and heterogeneous sample of biomedical editors and authors. Our plans for future work are to validate the ARCADIA tool.

**Acknowledgements** We thank Patrick Bossuyt, David Moher, Elizabeth Moylan, Albert Selva O'Callaghan, Alessandro Recchioni and José Jiménez Villa for piloting the survey questionnaire. We thank all people who participated in the survey.

**Contributors** All authors provided intellectual contributions to the development of this manuscript. CS, DH, IB and JAG jointly contributed to the study conception, design and interpretation of data. CS conducted the survey. CS, RR and JAG conducted the quantitative analysis and created the web application. CS, DB and DH conducted the qualitative analysis. CS, DH, AR, IB and JAG formed the steering committee. CS led the writing of the manuscript. IB and JAG led the supervision of the manuscript preparation. All authors provided detailed comments on earlier drafts and approved the final manuscript.

**Funding** This project was supported by the European Union's Horizon 2020 Research and Innovation Programme under the Marie Sklodowska-Curie grant agreement no 676207.

**Disclaimer** The funders had no role in the study design, data collection and analysis, decision to publish, or preparation of the manuscripts.

**Competing interests** None declared.

**Patient and public involvement** Patients and/or the public were not involved in the design, or conduct, or reporting, or dissemination plans of this research.

**Patient consent for publication** Not required.

**Ethics approval** The study was approved by the Research Committee of the Governing Council of the Universitat Politècnica de Catalunya, Barcelona Tech, Spain (Reference: EC 02, Date: 2 May 2018).

**Provenance and peer review** Not commissioned; externally peer reviewed.

**Data availability statement** Data are available in a public, open access repository. The dataset supporting the conclusions of the research reported in this paper will be available in the Zenodo repository in the Methods in Research on Research (MiRoR) community (https://zenodo.org/communities/miror/?page=1&size=20)

**ORCID iDs**
Cecilia Superchi http://orcid.org/0000-0002-5375-6018
Darko Hren http://orcid.org/0000-0001-6465-6568
David Blanco http://orcid.org/0000-0003-2961-9328

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
