## [Reviewer comments · BMJ Open]

ARTICLE DETAILS

TITLE (PROVISIONAL)	The development of ARCADIA: a tool for assessing the quality of peer review reports in biomedical research
AUTHORS	Superchi, Cecilia; Hren, Darko; Blanco, David; Rius, Roser; Recchioni, Alessandro; Boutron, Isabelle; González, José Antonio

VERSION 1 - REVIEW

REVIEWER	Armen Yuri Gasparyan Departments of Rheumatology and Research and Development, Dudley Group NHS Foundation Trust (Teaching Trust of the University of Birmingham, UK), Russells Hall Hospital, Dudley, West Midlands, UK
REVIEW RETURNED	22-Nov-2019

GENERAL COMMENTS	The study offers a new tool for evaluating reviewer comments. There are some optional comments. 1. As most participants were from Europe and North America (86.6%), is it possible to generalize the study conclusion?2. Which biomedical journals can benefit from the tool? Is it primarily for non-mainstream science journals struggling with the quality and indexability?3. Open-access journals tend to implement public peer review. Is there any specific approach to employ the ARCADIA for open-access journals?4. The survey respondents were editors and authors. It could be a limitation since it is unclear whether the same respondents are also active reviewers.
---

REVIEWER	Francisco Grimaldo University of Valencia Spain
REVIEW RETURNED	27-Jan-2020

GENERAL COMMENTS	This paper deals with an extremely important but not yet solved issue of assessing the quality of peer review reports. Publishers, editors, reviewers and authors are very much interested in this and the proposed checklist could become a key contribution. In fact, one of the first questions that come to mind is to which extent the defined domains and items are generalizable beyond the biomedical field. It would also be nice to have the authors opinion on
---

	how representative the sample of participants that completed the survey is. I wonder whether the authors could estimate the number of people reached and the success of the recruitment strategy. How many people reacted to the e-mail, was persuaded by "informants" or got to know about the survey on the Internet/Twitter? Was snowballing also applied when asking authors? More justification and discussion about the different demographic characteristics of editors and authors is needed. Characteristics can only be found when looking at the questionnaires in Supplementary File 1 (eg. occupation or type of institution might also work for editors). Table 2 and Supplementary File 4 are in part overlapped. It is weird that the majority of authors have +20 years of experience, while editors just show +5 years of experience. Is the profile of corresponding authors so biased towards senior researchers? Beyond descriptive statistics about the participants, I would expect the manuscript to include more data analysis on the effect of demographic characteristics on the responses. The web application made available to explore quantitative results is very nice and complete but I think that the paper should guide the reader and highlight the most relevant findings. For instance, the text lacks a discussion about the ranking of items sorted for editors and authors (eg. rank correlation), as well as about differences in outcomes depending on gender, location, job title, years of experience, type of institution, author-editor role, etc. The narrative approach followed to describe the activity of the steering committee makes it hard to keep track of how comments have been considered or discarded. The flowchart in Figure 1 does not give any hint about how items were merged, excluded or added when creating the survey and during the revision of the pilot questionnaire. Nor it does about how each response in the Supplementary Files 5 and 6 is (un)covered by the decisions of this committee (eg. sub-code "Leading to incorrect decision"). There might be a gap between how items are defined in Table 1 and presented to the participants (Supplementary File 1) and how they are finally converted into questions for the ARCADIA tool (Table 4). Could participants have answered differently if they had been asked about this precise wordings? Should the responses to this questions go beyond Yes/No/NA? May this be a potential limitation of the tool? What would happen if you used a scale instead of a checklist? Last but not least, what is the comparison between the proposed tool and the ones found in the previous systematic review? The authors state that "none reported any definition of peer review quality" but the dataset made available in the Zenodo repository (eg. ESR15_p1_DataExtraction_SR.xlsx) appears to include several papers with somehow overlapping list of items.
--	--

VERSION 1 – AUTHOR RESPONSE

Reviewer Name: Armen Yuri Gasparyan

Institution and Country: Departments of Rheumatology and Research and Development, Dudley Group NHS Foundation Trust (Teaching Trust of the University of Birmingham, UK), Russells Hall Hospital, Dudley, West Midlands, UK

Please state any competing interests or state 'None declared': None declared

Thank you very much for your valuable comments. Below you will find our responses to each of your points.

The study offers a new tool for evaluating reviewer comments.

There are some optional comments.

1. As most participants were from Europe and North America (86.6%), is it possible to generalize the study conclusion?

We appreciate your pointing that out. As you correctly reported, the study participants were mainly editors and authors from Europe and North America (337/389, 86.6%), which may limit the generalizability of the study results.

As we highlighted in the discussion of the manuscript, this result may be due to the recruitment strategy we used, especially to identify biomedical editors. Although we also utilized a snowballing strategy, we mainly contacted editors through ones of the main European biomedical publishers [i.e., the BMJ Publishing group and BMC (part of Springer Nature)]. Moreover, the survey was also circulated among the members of the European Association of Science Editors (EASE) through a newsletter.

However, according to the World RePORT (Research Portfolio Online Reporting Tools), an open access database of funding information developed by the US National Institutes of Health (NIH), research organizations from Europe and North America are those receiving more funding for health research (1) and consequently, where more research is conducted. In addition, the majority of biomedical journals are also located in Europe and North America. This is line with the sample of the survey participants of our study, which was mainly represented by editors and authors from Europe and North America.

Based on your comment, we have now expanded the limitations of the study in the discussion section incorporating a comment on the generalizability of the study conclusions (pag.14).

“Furthermore, the majority of editors and authors were from Europe and North America, which may limit the generalizability of the results. This result may be due to the recruitment strategy we used, especially to identify biomedical editors. Although we also utilized a snowballing strategy, we mainly contacted editors through European biomedical journals.”

We have also added this aspect as limitation in the “strengths and limitations” section after the abstract (pag.3).

“Strengths and limitations of this study

- ARCADIA constitutes the first tool that has been systematically developed to assess the quality of peer review reports.
- Its development is based on an exhaustive review of the literature and on empirical data from a large and heterogeneous sample of both biomedical editors and authors.
- The majority of editors and authors were from Europe and North America, which may limit the generalizability of the results.
- ARCADIA has not yet been validated.”

2. Which biomedical journals can benefit from the tool? Is it primarily for non-mainstream science journals struggling with the quality and indexability?

We are grateful for the constructive comment. We did not aim to address ARCADIA to any specific types of journals, since we believe that any biomedical journal could equally benefit from this checklist. This tool is particularly helpful to further increase the transparency of the entire peer review process. It could be used by editors from any biomedical journal to evaluate the reviewers' work and thus make an informed and unbiased decision on the submitted manuscript. In addition, ARCADIA could be also used by researchers as an outcome when evaluating interventions to improve the peer review process.

3. Open-access journals tend to implement public peer review. Is there any specific approach to employ the ARCADIA for open-access journals?

Thank you for this valid point. We believe that open access journals could publish the ARCADIA checklist along with the referees' reports. This way, ARCADIA would allow to further increase the transparency of the peer review process, by showing whether referees properly addressed all relevant aspects of the manuscript. Moreover, this checklist could also allow verifying if the editor's decision on such manuscript was based on high-quality peer review reports.

4. The survey respondents were editors and authors. It could be a limitation since it is unclear whether the same respondents are also active reviewers.

Thank you for this valuable comment. We do not believe that the overlapping of different roles related to the peer review process, among survey participants, could be a limitation of the study. Contrary, we believe that it could be a possible strength, since the duplicity of roles of some participants could bring a more complete perspective on the importance and complexity of assessing peer review report quality in biomedical research.

References

1. NIH. World RePORT [Internet]. Available from: <https://worldreport.nih.gov/app/#!/>

Reviewer Name: Francisco Grimaldo

Institution and Country: University of Valencia, Spain

Please state any competing interests or state 'None declared': None declared

We are extremely thankful for the constructive comments. We very much appreciate the positive observations as well as notes on potential issues. Below you will find our responses to each of your points.

This paper deals with an extremely important but not yet solved issue of assessing the quality of peer review reports. Publishers, editors, reviewers and authors are very much interested in this and the proposed checklist could become a key contribution. In fact, one of the first questions that come to mind is to which extent the defined domains and items are generalizable beyond the biomedical field. It would also be nice to have the authors opinion on how representative the sample of participants that completed the survey is. I wonder whether the authors could estimate the number of people reached and the success of the recruitment strategy. How many people reacted to the e-mail, was persuaded by "informants" or got to know about the survey on the Internet/Twitter? Was snowballing also applied when asking authors?

Thank you for this valuable comment. ARCADIA is an evidence-based tool that it was systematically developed using evidence exclusively related to the biomedical research. Therefore, we believe that

the checklist, as it is, should not be applied to other fields. However, it could be used as a model, or starting point to develop similar tools in other disciplines.

Regarding the representativeness of the sample, the study participants were mainly from Europe and North America (337/389, 86.6%). Particularly, the majority of biomedical editors (132/165, 80%) who took part in the survey were from Europe.

As we highlighted in the discussion of the manuscript, this result may be due to the recruitment strategy we used, especially to identify biomedical editors. Although we also utilized a snowballing strategy, we mainly contacted editors through ones of the main European biomedical publishers [i.e., the BMJ Publishing group and BMC (part of Springer Nature)]. Moreover, the survey was also promoted among the members of the European Association of Science Editors (EASE) through newsletter.

However, according to the World RePORT (Research Portfolio Online Reporting Tools), an open access database of funding information developed by the US National Institutes of Health (NIH), research organizations from Europe and North America are those receiving more funding for health research (1) and consequently, where more research is conducted. In addition, the majority of biomedical journals are also located in Europe and North America. This is line with the sample of the survey participants of our study, which was mainly represented by editors and authors from Europe and North America. Based on your comment, we now expanded the limitations of the study in the discussion section commenting on the generalizability of the study (pag.14).

“Furthermore, the majority of editors and authors were from Europe and North America, which may limit the generalizability of the results. This result may be due to the recruitment strategy we used, especially to identify biomedical editors. Although we also utilized a snowballing strategy, we mainly contacted editors through European biomedical journals.”

Based on your suggestions, we have now added some information in the manuscript on the success of the recruitment strategy (pag.8).

“Of the 1134-biomedical editors and 3633 corresponding authors invited via email, 89 (7.8%) and 238 (6.5%) completed the survey, respectively. In addition, 109 editors and 10 authors completed the survey using the web link.”

The snowballing strategy was used to identify only biomedical editors as we reported in the manuscript (pag.6).

“In the invitation email and newsletter, the editors were encouraged to forward the survey to colleagues who might be interested in issues related to peer review. This recruitment strategy, known as snowballing, allowed us to identify “information-rich key informants” among biomedical editors [8].”

However, since the survey was promoted and re-shared on Twitter by different users providing not always thorough instructions, we included in the first page of the survey directed to biomedical editors, also the link to the questionnaire addressed to authors. This was done in case a researcher, which was not an editor and erroneously opened the link to the survey questionnaire, was still able to participate to the study as biomedical author. However, only 10 authors completed the survey using the web link. We have now added this aspect into the manuscript (pag.7)

“We did not use a snowballing strategy to recruit authors. However, since the survey directed to biomedical editors was promoted on Twitter by different users who sometimes did not provide thorough instructions, we included in the first page of the survey, also the link to the questionnaire

addressed to authors. This was done so that a researcher, who was not an editor and mistakenly opened the link to the survey questionnaire, was still able to participate to the study as biomedical author.”

More justification and discussion about the different demographic characteristics of editors and authors is needed. Characteristics can only be found when looking at the questionnaires in Supplementary File 1 (eg. occupation or type of institution might also work for editors). Table 2 and Supplementary File 4 are in part overlapped. It is weird that the majority of authors have +20 years of experience, while editors just show +5 years of experience. Is the profile of corresponding authors so biased towards senior researchers?

We appreciate your pointing that out. Based on your suggestions, we have now added more information in the manuscript on the demographics characteristics of both editors and authors (pag.9).

“Editors were mainly associate editors (63/165, 38.2%) and editors in chief (350/165 0.3%), primarily involved in making decisions on the submitted manuscripts (144/165, 87.3%). Most of them worked in specialty journals (126/165, 76.4%) and had experience as authors of scientific papers (141/165, 85.5%). The corresponding authors were mainly professors (63/224, 28.1%), but also PhD students, postdocs or lecturers (49/224, 21.9%) or researchers (47/224, 21%). Most of them worked in public universities (134/224, 59.8%) and were not employed as editor (161/224, 71.9%) in biomedical journals. Among those who also worked as biomedical editors (63/224, 28.1%), 88.9% were involved in making decision on the manuscript (Supplementary file 4).”

In addition, we removed the overlapping information in the Supplementary file 4.

Finally, we believe that the fact that the majority of corresponding authors were senior researchers is due to the previously described tendency of confusing corresponding authorship with author seniority (2).

Beyond descriptive statistics about the participants, I would expect the manuscript to include more data analysis on the effect of demographic characteristics on the responses. The web application made available to explore quantitative results is very nice and complete but I think that the paper should guide the reader and highlight the most relevant findings. For instance, the text lacks a discussion about the ranking of items sorted for editors and authors (eg. rank correlation), as well as about differences in outcomes depending on gender, location, job title, years of experience, type of institution, author-editor role, etc.

We are grateful for the constructive comment. Based on your suggestions, we have now added more findings related to the effect of demographic characteristics on the responses by conducting PCA. Due to the word limit of the journal, we have only added the most relevant results (pag.10).

“The study of the supplementary variables did not reveal any differences between authors and editors in terms of item rating. However, we found that female participants above the age of 55 generally provided higher rating for the items, compared to younger male participants.”

In addition, we have now added more discussion on the ranking of items sorted for editors and authors (pag.14).

“Overall, we did not find any differences between authors and editors in terms of item rating by conducting PCA. Only two items, Timeliness and Detail/thoroughness, presented a difference according to the separate mean score rankings of authors and editors. Timeliness was considered

more important for authors and this could be justified by the fact that authors are usually more interested in receiving decisions about their manuscript as soon as possible. However, editors rated detail/thoroughness as more important to them, given thorough and detailed peer review reports help them make a better editorial decision on any given manuscript.”

The narrative approach followed to describe the activity of the steering committee makes it hard to keep track of how comments have been considered or discarded. The flowchart in Figure 1 does not give any hint about how items were merged, excluded or added when creating the survey and during the revision of the pilot questionnaire. Nor it does about how each response in the Supplementary Files 5 and 6 is (un)covered by the decisions of this committee (eg. sub-code "Leading to incorrect decision").

This is a relevant comment. We have now updated the text providing more information on how items were merged, excluded or added when creating the survey (pag.5).

“A systematic review allowed the identification of 24 tools, aimed at assessing the quality of peer review reports [6]. We extracted 132 items from such tools. After removing the redundant items, we obtained 17 items. We then eliminated two items and incorporated five new ones that met our definition of peer review report quality, after piloting the survey questionnaire and discussing with the steering committee. Overall, 20 items were identified to assess peer review report quality (Table 1).”

We have also updated Figure 1.

There might be a gap between how items are defined in Table 1 and presented to the participants (Supplementary File 1) and how they are finally converted into questions for the ARCADIA tool (Table 4). Could participants have answered differently if they had been asked about this precise wordings? Should the responses to this questions go beyond Yes/No/NA? May this be a potential limitation of the tool? What would happen if you used a scale instead of a checklist?

Thank you for highlighting this. The items reported in Table 1 are the same as those reported in the survey questionnaire. In the ARCADIA checklist, we decided to report the items as questions, because we concluded they were more straightforward to answer. In addition, the items included in ARCADIA are worded differently, compared to those included in the survey questionnaire. This is for we took into account the comments provided by the survey participants and steering committee, to formulate the final items to include in the ARCADIA checklist.

Regarding the ARCADIA response options (Yes/No/NA) and the use of a scale instead of a checklist as possible limitations of the tool, you made extremely valid points that are beyond the scope of this manuscript. We do, however, address all these points (or limitations) in our next paper about the validation process of the ARCADIA tool.

As we reported in our previous methodological systematic review (3), we believe that checklists may be more appropriate means to assess quality for several reasons. By definition, a scale “combines information on several features in a single numerical score” while, a checklist “examines key dimensions individually” (4). In other words, a scale includes an overall score, while a checklist does not. The use of overall quality scores is still under debate (4-6). In particular, overall quality scores do not take into account that the importance of individual items varies between contexts. For instance, some biomedical journals prioritize the originality of the manuscripts, while others the strength and validity of the methods. Therefore, we decided to develop a checklist of items that could be used to show the independent influence of each item on the overall peer review report quality (5).

In addition, since checklists do not present an overall score, they do not require a weight for the items. It has been shown that “choices on how to weight and calculate quality scores are generally arbitrary, thus it would be impossible to produce an objective quality score” (6). Therefore, each ARCADIA item should be ticked just as ‘Yes’, ‘No’ or ‘NA’.

ARCADIA does not ask the user to rate the quality of peer review reports, but presents a list of quality items that should appear in a peer review report, to help editors make an informed and unbiased decision about the study’s outcome and aid authors to improve the quality of their manuscript.

Last but not least, what is the comparison between the proposed tool and the ones found in the previous systematic review? The authors state that “none reported any definition of peer review quality” but the dataset made available in the Zenodo repository (eg. ESR15_p1_DataExtraction_SR.xlsx) appears to include several papers with somehow overlapping list of items.

We are grateful for the constructive comment. As you correctly noted, we found that none of the tools identified in our methodological systematic review provided a definition of quality of peer review reports. Moreover, we found that only one tool described the scale development. The first version of this tool was designed by four researchers and three editors and it was based on a tool used in an earlier study and that had been developed by reviewing the literature and interviewing editors. However, no information was provided on how this previous tool was actually developed.

Compared to other tools, ARCADIA is the first one systematically developed to assess the quality of peer review reports. In addition, its development is based on an exhaustive review of the literature and on empirical data from a large and heterogeneous sample of both biomedical editors and authors.

In the methodological systematic review, we also found that a total number of 18 tools had multiple items, which help the users to assess the quality of peer review reports. To identify the items to be included in ARCADIA, we first extracted all items from these 18 tools, merged the redundant ones and included only those items that met our definition of peer review report quality, as we reported more in the detail in the manuscript. This is the reason why some items included in ARCADIA appear similar to those identified with the systematic review.

References

1. NIH. World RePORT [Internet]. Available from: <https://worldreport.nih.gov/app#!/>
2. Zauner H, Nogoy NA, Edmunds SC, Zhou H, Goodman L. Editorial: We need to talk about authorship. *GigaScience* [Internet]. 2018 Dec 1 [cited 2020 Apr 13];7(12). Available from: <https://academic.oup.com/gigascience/article/doi/10.1093/gigascience/giy122/5114264>
3. Superchi C, González JA, Solà I, Cobo E, Hren D, Boutron I. Tools used to assess the quality of peer review reports: a methodological systematic review. *BMC Med Res Methodol*. 2019;19(1):48.
4. Jüni P, Altman DG, Egger M. Systematic reviews in health care: assessing the quality of controlled clinical trials. *BMJ* [Internet]. 2001 [cited 2011 Jan 21]; 323 (7303): 42-6.
5. Boutron I, Moher D, Tugwell P, Giraudeau B, Poiraudeau S, Nizard R, et al. A checklist to evaluate a report of a nonpharmacological trial (CLEAR NPT) was developed using consensus. *J Clin Epidemiol*. 2005 Dec;58(12):1233–40.
6. Whiting P, Rutjes AW, Reitsma JB, Bossuyt PM, Kleijnen J. The development of QUADAS: a tool for the quality assessment of studies of diagnostic accuracy included in systematic reviews. *BMC medical research methodology*. 2003 Dec 1;3(1):25.

VERSION 2 – REVIEW

REVIEWER	Francisco Grimaldo University of Valencia, Spain
REVIEW RETURNED	12-May-2020

GENERAL COMMENTS	The authors have addressed all my previous concerns. I agree with the responses and I will be happy to see this paper published.
--